# Whole-Genome Characterisation of ESBL-Producing *E. coli* Isolated from Drinking Water and Dog Faeces from Rural Andean Households in Peru

**DOI:** 10.3390/antibiotics11050692

**Published:** 2022-05-20

**Authors:** Maria Luisa Medina-Pizzali, Apoorva Venkatesh, Maribel Riveros, Diego Cuicapuza, Gabriela Salmon-Mulanovich, Daniel Mäusezahl, Stella M. Hartinger

**Affiliations:** 1Research Unit in Integral Development, Environment and Health, School of Public Health and Administration, Universidad Peruana Cayetano Heredia, Lima 15102, Peru; maria.medina.p@upch.pe (M.L.M.-P.); gsalmonm@pucp.edu.pe (G.S.-M.); daniel.maeusezahl@unibas.ch (D.M.); 2Department of Epidemiology and Public Health, Swiss Tropical and Public Health Institute, 4123 Allschwil, Switzerland; apoorva.venkatesh@swisstph.ch; 3University of Basel, 4001 Basel, Switzerland; 4School of Medicine, Universidad Peruana Cayetano Heredia, Lima 15102, Peru; maribel.riveros@upch.pe; 5Laboratory of Microbial Genomics, Department of Cellular and Molecular Sciences, School of Science and Philosophy, Universidad Peruana Cayetano Heredia, Lima 15102, Peru; diego.cuicapuza@upch.pe; 6Institute for Earth, Nature and Energy, Pontificia Universidad Católica del Perú, Lima 15088, Peru

**Keywords:** phylogenomic analysis, one health, ESBL-producing *E. coli*, carbapenem resistance, whole-genome sequencing

## Abstract

*E. coli* that produce extended-spectrum β-lactamases (ESBLs) are major multidrug-resistant bacteria. In Peru, only a few reports have characterised the whole genome of ESBL enterobacteria. We aimed to confirm the identity and antimicrobial resistance (AMR) profile of two ESBL isolates from dog faeces and drinking water of rural Andean households and determine serotype, phylogroup, sequence type (ST)/clonal complex (CC), pathogenicity, virulence genes, ESBL genes, and their plasmids. To confirm the identity and AMR profiles, we used the VITEK^®^2 system. Whole-genome sequencing (WGS) and bioinformatics analysis were performed subsequently. Both isolates were identified as *E. coli*, with serotypes -:H46 and O9:H10, phylogroups E and A, and ST/CC 5259/- and 227/10, respectively. The isolates were ESBL-producing, carbapenem-resistant, and not harbouring carbapenemase-encoding genes. Isolate 1143 ST5259 harboured the *astA* gene, encoding the EAST_1_ heat-stable toxin. Both genomes carried ESBL genes (*bla*_EC-15_, *bla*_CTX-M-8_, and *bla*_CTX-M-55_). Nine plasmids were detected, namely IncR, IncFIC(FII), IncI, IncFIB(AP001918), Col(pHAD28), IncFII, IncFII(pHN7A8), IncI1, and IncFIB(AP001918). Finding these potentially pathogenic bacteria is worrisome given their sources and highlights the importance of One-Health research efforts in remote Andean communities.

## 1. Introduction

Antimicrobial resistance (AMR)—particularly in the *Enterobacteriaceae* family—has become a problem of great relevance worldwide due to its increasing prevalence, and the emergence of multiple-drug-resistant strains AMR can affect everyone, irrespective of age. It is estimated that, in 2019, 4.95 million people died from illnesses associated with bacterial AMR. Of those, 1.27 million deaths, mostly in low- and middle-income countries (LMICs), were directly attributable to bacterial AMR [1]. Enterobacteria are the most important etiological agents of serious hospital-acquired and community-onset bacterial infections in humans [2,3,4,5]. In South America, resistance to β-lactam antibiotics and fluoroquinolones is a major problem when treating enterobacterial infections [6]. In addition, reports of the emergence of colistin resistance in this region have been recently published [7,8]. South American countries show some of the highest rates of AMR in *Enterobacteriaceae* worldwide. Huge socioeconomic differences, broad access to antimicrobials, and ineffective health systems and sanitation problems favour the emergence and spread of resistant bacterial strains [6]. In this region, antibiotic use is widespread in human medicine and animal production; thus antibiotic residues and resistant enterobacteria and AMR genes have been detected in soil, water from different environs, agricultural products (produce), and livestock. In rural Latin America, anthropogenic activities such as animal husbandry, fish farming, and agriculture were identified as drivers for antimicrobial resistance dissemination [9]. The most frequent animal contributions were the carriage and/or transfer of antimicrobial resistance determinants, the inadequate or unregulated use of veterinary antimicrobials, and the spread of resistant bacteria throughout the food supply chain via foods of animal origin [10]. Water was the most frequently identified among the environmental contributors to AMR spread in Andean Peru [9,11,12].

β-Lactamases are the leading cause of resistance to β-lactam antibiotics, and β-lactamase production is the primary mechanism of antibiotic resistance in *Escherichia coli* [13]. Among β-lactamases, extended-spectrum β-lactamases (ESBLs) that mediate resistance to all penicillins, third-generation cephalosporins, and aztreonam are of utmost importance, given that they substantially reduce the available antibiotic treatment options [14]. ESBLs include the extended-spectrum TEM-, SHV-, OXA-, and cefotaxime (CTX)-M type enzymes [15], while *E. coli* that produce CTX-Ms are most commonly associated with ESBLs [16]. AmpC β-lactamases —which function as cephalosporinases— are also clinically important. They hydrolyse cephamycins and other extended-spectrum cephalosporins and are poorly inhibited by clavulanic acid; in some cases, they confer resistance to carbapenems [17].

In Peru, previous studies have investigated ESBLs in enterobacteria isolated from clinical, community, and peri-urban settings. However, few reports have characterised ESBL-producing enterobacteria at the molecular level, and if so, mostly in clinical settings [18,19,20,21,22]. Using molecular methods, we previously characterised ESBL-producing enterobacteria from rural Andean communities [12,23]. However, literature on genomic characterisation of ESBL bacteria in Peru is very scarce [24,25,26]. The present study is the first report of the whole-genome characterisation of ESBL-producing *E. coli* isolates from water and farm/companion animals in a rural Andean setting in Peru. This study aims to confirm the identity and AMR profile and determine the serotype, phylogroup, sequence type (ST) and clonal complex (CC), pathogenicity, virulence genes, ESBL genes, and type of plasmids for ESBL-producing *E. coli* isolates.

## 2. Materials and Methods

### 2.1. Study Setting and Design

Hartinger et al. studied the dissemination pathways of AMR in humans, animals, and the environment in the Cajamarca Region in the northern highlands of Peru. The cross-sectional study design and sampling scheme are described elsewhere [23]. In brief, they sampled 40 households. Their study collected two rectal or cloacal swabs of fresh stool samples from a domestic animal (dog, cat) and/or a farm animal (cow, pig, fowl) from each household (*n* = 80). Additionally, two water samples per household were collected, one from the household’s drinking water (DW) source and/or one from the household’s primary water source (*n* = 80). The two ESBL-positive isolates analysed in the present study originated from two out of 40 households sampled. They were isolated from one water sample and one dog faeces sample, as shown in Appendix A.

### 2.2. Households’ Characteristics

Hartinger et al. found only one ESBL isolate from a dog and an ESBL isolate from the DW source from two homes [23]. In these households, animals were allowed to roam freely in the kitchen and courtyard, and their faeces were found in both areas. Both homes used piped water from the public water network system but had no sewage system, as they had installed latrines instead.

### 2.3. Microbiological Identification and Antimicrobial Susceptibility Testing

The microbiological identification and the antimicrobial susceptibility testing were performed using a VITEK^®^ 2 Compact automated system (bioMérieux, Marcy l’Etoile, France). A colorimetric reagent card (GN) to identify the most significant fermenting and non-fermenting Gram-negative bacilli was used according to the manufacturer’s instructions. AST-N249 cards were prepared according to the manufacturer’s instructions. The minimum inhibitory concentrations for the following antimicrobials were recorded (ampicillin/sulbactam (SAM), piperacillin/tazobactam (TZP), cefazolin (CFZ), cefuroxime (CXM), cefotaxime (CTX), ceftazidime (CAZ), cefepime (FEP), ertapenem (ETP), imipenem (IPM), meropenem (MEM), amikacin (AMK), gentamicin (GEN), ciprofloxacin (CIP), colistin (CST), and tigecycline (TGC). Previously [23], antimicrobial susceptibility testing for other antimicrobials has been performed using commercial discs [27]; *Escherichia coli* ATCC 25922 was used as a reference strain for quality assurance of the susceptibility testing of *E. coli*.

### 2.4. Phenotypical and Molecular Identification of ESBL-Producing Ability

The phenotypical ESBL-producing ability of the isolates was determined using the Jarlier method [28] for the following antibiotics: aztreonam (5 µg disk), ceftazidime (30 µg disk), cefotaxime (30 µg disk), ceftriaxone (30 µg disk), amoxicillin/clavulanic acid (30 µg disk) and cefepime (30 µg disk), and confirmed by the combined disc method [27]. Further molecular confirmation was performed using conventional polymerase chain reaction PCR. These procedures have been described elsewhere [23]. Briefly, genes encoding TEM-, SHV-, OXA-, and CTX-M-type β-lactamases (*bla*_TEM_, *bla*_SHV_, *bla*_OXA_, *bla*_CTX-M-U_, *bla*_CTX-M-2_, *bla*_CTX-M-3_, *bla*_CTX-M-8_, *bla*_CTX-M-9_, *bla*_CTX-M-10_) were PCR-amplified using the primers shown previously [23].

### 2.5. Genome Sequencing and Bioinformatic Analysis

The genomic DNA was extracted from overnight cultures using the GeneJET genomic DNA purification kit (Thermo Fisher Scientific, Waltham, MA, USA), according to the manufacturer’s instructions. DNA concentration was evaluated by Qubit^®^ 4.0 fluorometer (Life Technologies, Carlsbad, CA, USA). The genomic library was constructed using a Nextera XT DNA Library Preparation Kit (Illumina, Inc., San Diego, CA, USA) and subsequently sequenced using a 2 × 250 paired-end library on a MiSeq platform (Illumina). The quality control of each sequence was evaluated using Fastqc v0.11.5 [29], and Trimmomatic v0.38 [30] was used to remove adapters and filter low-quality reads. The reads were assembled de novo using SPAdes v.3.15.2 [31]. The bioinformatic analysis was carried out using the tools available at the Center of Genomic Epidemiology (www.genomicepidemiology.org) (accessed on 1 October 2021), setting identity at ≥90% and coverage at ≥90%. We performed *E. coli* serotyping (SerotypeFinder v.2.0), in silico phylogenetic typing (ClermonTyping (https://github.com/A-BN/ClermonTyping) (accessed on 1 January 2022), multilocus sequence typing (MLST v.2.0), pathogenicity (Pathogen Finder v.1.1), virulence genes detection (VirulenceFinder v.2.0), plasmid replicon typing (PlasmidFinder v.2.1 and MobileElementFinder v.1.0), and ESBL genes detection using ABRicate v.1.0.1 (https://github.com/tseemann/abricate) (accessed on 1 January 2022) NCBI and ResFinder databases. Raw Illumina reads were uploaded to GenBank under BioProject PRJNA816508.The complete outputs for all the results of the bioinformatic tools are shown in Appendix A.

## 3. Results

### 3.1. Microbiological Identification and Antimicrobial Susceptibility Testing of the E. coli Isolates

For this analysis, only ESBL-phenotypically-positive isolates obtained in the mother study were included, one from dog faeces (Isolate 1143) and one from the household’s DW source (Isolate 1144). Isolates 1143 and 1144 were identified as *E. coli* strains with a 93% and 96% probability using VITEK^®^2 (bioMérieux, Marcy l’Etoile, France) automated reading and further confirmed by WGS.

Based on the VITEK^®^2 system, both isolates were resistant to SAM (combined effect); cephalosporins CFZ, CXM, and CTX; TZP (β-lactam/β-lactamase inhibitor combination); and MEM (a carbapenem). However, only Isolate 1143 was resistant to important cephalosporins CAZ and FEP and the combination antibiotic SXT. On the other hand, only Isolate 1144 was resistant to GEN and CIP, while 1143 was not. We compared the VITEK^®^2 automated readings for antimicrobial susceptibility to the disk diffusion assay results, as shown in Appendix A. It is noteworthy that Isolate 1144 showed resistance to FEP by the disk diffusion method, contrary to the results obtained using the automated VITEK^®^2 platform. The VITEK^®^2 system identified Isolate 1143 as an ESBL producer. Isolate 1144, however, was characterised as ESBL negative based on the MIC results, which were interpreted automatically by the system based on the VITEK^®^2 breakpoints (see Appendix A).

### 3.2. Detection of ESBL-Encoding Genes by PCR and Whole-Genome Sequencing Analysis (WGS)

Detection of the most common β-lactamase-encoding genes (*bla*_TEM_, *bla*_SHV_, *bla*_OXA_, and *bla*_CTX-M_) was carried out using PCR. None of the isolates showed amplification for *bla*_SHV_, and *bla*_OXA_ genes, but both harboured *bla*_TEM_ and *bla*_CTX-M_ genes. WGS analysis using Resfinder and NCBI identified the presence of genes encoding resistance to cephalosporins (*bla*_CTX-M-8_, *bla*_CTX-M-55_, *bla*_EC-15_), broad-spectrum β-lactamases (*bla*_TEM-1_), tetracyclines (*tet*(B), *tet*(M)), aminoglycosides (*aac(3)-Iid*, *aph(3’)-Iia*, *aph(3’’)-Ib*, *aph(6)-Id*, *aadA1* and *aadA2*), sulphonamides (*sul3*), phenicols (*floR*, *cmlA1*), fosfomycin (*fosA3*), quinolone (*qnrB19*), and trimethoprim (*dfrA12*), as shown in Table 1. The genotypic resistance of both isolates correlated with their phenotypical resistance, except for MEM; the resistance to this carbapenem was only phenotypically observed.

Genomic analysis using MLST 2.0 indicated that the *E. coli* Isolates 1143 and 1144 belonged to ST5259 and ST227, respectively, and 1144 belonged to CC 10. According to the genomes’ analysis using the ClermontTyping tool, the 1143 and 1144 isolates corresponded to the phylogenetic groups E and A, respectively. Using SerotypeFinder v.2.0, Isolate 1143 was classified as serotype -:H46, whereas Isolate 1144 belonged to serotype O9:H10, as presented in Table 1.

### 3.3. Pathogenicity, Virulence Genes, and Type of Plasmids

Using Pathogenfinder 1.1 and VirulenceFinder 2.0, both isolates were identified as likely human pathogens, but Shiga-toxin genes were not detected in their genomes. A total of 28 different virulence genes were detected from the *E. coli* genomes. The colicin ia (*cia*) virulence gene was detected in both *E. coli* isolates. Other prevalent virulent genes included: *astA*, *iroN*, *chuA*, *gad*, *ompT*, *capU*, *cea*, *fyuA*, *sitA*, *irp2 iutA*, and *terC*. The *astA* gene (gene product: EAST_1_ heat-stable toxin) was found in Isolate 1143 only.

Using the PlasmidFinder 2.1 and the MobileElementFinder 1.0 tools, four plasmids were detected for Isolate 1143, IncR (harbouring the *aph(3’’)-Ib* and *aph(6)-Id* aminoglycoside resistance genes), IncFIC(FII) (harbouring the florfenicol/chloramphenicol resistance *floR* gene), IncI1 (carrying the *cia* virulence gene), and IncFIB(AP001918). On the other hand, for Isolate 1144, five plasmids were identified, Col(pHAD28) (harbouring the *qnrB19* ciproflaxacin resistance gene), IncFII, IncFII(pHN7A8), IncI1, and IncFIB(AP001918).

## 4. Discussion

We confirmed the identity of the two enterobacteria isolates (1143 and 1144) as *E. coli*; Isolate 1143—obtained from dog faeces—was an atypical non-lactose fermenter. Lactose-negative uropathogenic *E. coli* strains have been recently isolated from dogs in Brazil [32], and they cause urinary tract infections in humans [33]. Further, lactose-negative *E. coli* are usually enteroinvasive (EIEC), are closely related to *Shigella* spp., and produce a dysenteric form of diarrhoea in humans [34]. Both isolates were likely human pathogens, with serotypes -:H46 and O9:H10, respectively. The *E. coli* O9:H10 serotype has been listed among the typical enteroaggregative (EAEC) *E. coli* serotypes in Brazil [35], and the same serotype was identified among samples of Shiga-toxin-producing *E. coli* (STEC) and Enteroaggregative *E. coli* (EAEC) from Mexico [36].

Both isolates were confirmed as ESBL producers given their genomes encoded ESBL genes (*bla*_CTX-M-8,_ *bla*_CTX-M-55_, *bla*_EC-15_), contrary to the VITEK^®^2 automatised system’s classification of Isolate 1144 as a non-ESBL producer. The VITEK^®^2 ESBL test has a varying sensitivity for detecting ESBL-producing enterobacteria correctly; it ranges between 86% and 98.1% [37,38]. The VITEK^®^2 AST N249 card used in our study identified Isolate 1143 as an ESBL producer, but not Isolate 1144, since it was not resistant to FEP and CAZ. However, Isolate 1144 showed a phenotypical resistance to other oxyimino cephalosporins CXM and CTX.

Isolate 1143 and Isolate 1144 belonged to phylogroups E and A, respectively. Phylogroup E includes the O157:H7 EHEC lineage, which is only a small subset of the whole genetic diversity found within the group. Phylogroup-E strains are rarely isolated compared to other phylogroup strains. They have been isolated from animals, the environment, and humans, and virulent/resistant strains are linked to humans [39]. Thus, finding an ESBL-producing, potentially pathogenic, phylogroup-E isolate in dog faeces points to a probable transmission route from humans. On the other hand, most phylogroup-A strains are human commensals and can be found in wastewater [40] and animals, especially poultry [41]. Notably, in our setting, backyard chicken farming is widespread (83% of households), and farm animals, in general, are administered antibiotics by the owners without prescription nor technical supervision (unpublished data from our research group). Other studies carried out in rural locations in Peru show similar findings [42,43]. Chickens roam freely, and household members are in contact with their animals and their scattered faeces. On the other hand, the piped water supply and the soil near the houses were contaminated with faecal coliforms originating from the environment, humans, or animals [23]. The fact that both isolates in this study displayed resistance to carbapenems is of great concern, given that this “third line” antimicrobial medication is exclusively for human use [44]. Thus, the ESBL/AmpC carbapenem-resistant *E. coli* isolates found in DW and dog faeces could originate from other farm animals or humans and be transmitted by different pathways, as shown by Hartinger et al. in our study area [23].

Isolates 1143 and 1144 belonged to ST/CC 5259/- and 227/10, respectively. We found that *E. coli* ST227 are considerably common and have been frequently reported around the world. In the public database EnteroBase (https://enterobase.warwick.ac.uk) (accessed on 1 February 2022), we found 84 publicly available *E. coli* ST227 genome assemblies from human, animal, and environmental sources. In Tunisia, an ST227 carbapenem-resistant *E. coli* isolate of human origin was reported in 2016, carrying the *bla*_OXA-48_, *bla*_CTX-M-15_, *bla*_TEM_, and *bla*_OXA-1_ genes [45]. A carbapenem-resistant *E. coli* isolate of the ST227 lineage was isolated from clinical samples in Lebanon, carrying the *bla*_OXA-48_ and the *bla*_TEM-1_ genes [46]. *E. coli* ST5259 are scarce; only two were found in EnteroBase, one of them originated from poultry in Ecuador and the other from humans in the United States of America. In China, an *E. coli* ST5259 genome of human origin was mentioned as part of BioProject PRJNA400107, being reported along with other *E. coli* genomes as carriers of the *mcr-1* gene, which is a plasmid-mediated colistin resistance gene [47].

Many of the identified virulence genes were associated with iron acquisition systems, and a small part was associated with toxin production, metal resistance (tellurite), and other virulence determinants, as shown in Appendix A. The *cia* gene—involved in killing other bacteria—was found in both isolates. The iron uptake genes detected were: *iroN*, *chuA*, *fyuA*, *iutA*, and *sitA*. The *chuA* gene encodes an outer membrane hemin receptor, and the *fyuA* gene encodes the yersiniabactin receptor, and both contribute most during infection of the urinary tract by uropathogenic *E. coli* [48]. The *chuA* gene was detected in Isolate 1143, whereas the *fyuA* gene was found in Isolate 1144 along with the other iron acquisition genes. This flags Isolate 1144 as a possible uropathogenic strain, given it harboured many genes that enable iron uptake, best suited for iron-deprived environments such as human urine [49]. On the other hand, Isolate 1143 harboured the *astA* gene, encoding the EAST_1_ heat-stable toxin, which is associated with enteroaggregative *E. coli* (EAEC) in humans and enterotoxigenic *E. coli* (ETEC) in porcines [50]. None of the isolates’ genomes harboured Shiga-toxin genes.

The *bla*_CTX-M-55_ is one of the most abundant ESBL genes in the *Enterobacteriaceae* family, with a rising prevalence in *E. coli* from livestock and humans, especially in China [51,52]. It is usually carried by plasmids, but it has also been found chromosomally [52]. In Canada, enterobacteria isolated from turkeys carried the *bla*_CTX-M-55_ gene mediated by *IncF* plasmids [53]. In Japan, the *bla*_CTX-M-8_ gene was found in *E. coli* from humans and retail chicken meat, and its transmission was associated with IncI1 plasmids [54].

Unlike *bla*_CTX-M-55_ and *bla*_CTX-M-8_, which code for Class A ESBL, the *bla*_EC-15_ gene product belongs to the Class C β-lactamases [55], also known as AmpC-type β-lactamases. In comparison to ESBL (including Class A β-lactamases), AmpC-type β-lactamases hydrolyse broad- and extended-spectrum cephalosporins (cephamycins as well as oxyimino-β-lactams) but are not inhibited by clavulanic acid or other β-lactamase inhibitors. EC β-lactamases—encoded in *bla*_EC_ genes—are a specific type of AmpC β-lactamases found in *E. coli* [56]. In a recent One-Health study in Canada, the *bla*_EC-15_ gene was detected in *E. coli* isolates from different human, animal, and environmental sources. It did not occur in all isolates, having a prevalence of 16% [57] The fact that both isolates in our study carried genes for Class A ESBL and AmpC β-lactamases has great clinical importance, given that AmpC/ESBL *E. coli* cannot be treated with clavulanic acid, which complicates the treatment [58]. However, we did not include a cephalosporin/clavulanic acid combination in the disc diffusion assay, so resistance to clavulanic acid was not phenotypically confirmed. Further, both isolates expressed resistance to other antibiotics such as fluoroquinolones and a carbapenem, narrowing treatment options even further. According to Guzmán-Blanco et al. [59], in Latin America, the rates of nosocomial infection caused by ESBL-producing enterobacteria—especially CTX-M enzyme producers—are higher than in other regions. A One-Health-based study in Brazil—the biggest country in the region—showed that ESBL-producing *E*. *coli* carrying a diversity of *bla_CTX_*_-M_ gene variants and *mcr-1* genes are endemic across their territory at the interface between humans and animals [60]. ESBL-producing enterobacteria can also be resistant to cefepime and exhibit resistance to fluoroquinolones and ampicillin/sulbactam, as well as to aminoglycosides and piperacillin/tazobactam. Fortunately, more than 90% of ESBL-producing enterobacteria are still susceptible to carbapenems [59], although reports of carbapenem-resistant enterobacteria alarmingly increase in Latin American countries [61].

Despite the phenotypic resistance to meropenem of both isolates in our study, we did not find evidence of well-known carbapenemase-encoding genes—*bla*_KPC_, *bla*_NDM_, *bla*_NDM_, *bla*_IMP_, *bla*_VIM_ and *bla*_OXA-48_—or their variants [62]. Thus, non-carbapenemase mechanisms could be suspected for the isolates in our study. For instance, point mutations in porin genes reduce membrane permeability to carbapenems, in combination with hyperproduction of the β-lactamase AmpC or ESBL. Therefore, the isolates could be characterised as non-carbapenemase, carbapenem-resistant *E. coli* [63,64]. In Singapore, the emergence of this type of bacteria in clinical settings was suggested to result from selective antibiotic pressure along with *de novo* mutations or genetic reassortment in carbapenem-sensitive enterobacteria [65].

Plasmids play a major role in the dissemination of AMR genes in Gram-negative bacteria. Usually, they harbour multiple physically connected genetic determinants, conferring resistance to different classes of antibiotics simultaneously, such as extended-spectrum β-lactams, carbapenems aminoglycosides, sulfonamides, and quinolones [66,67]. Plasmid families, including IncF, IncI1, IncI2, IncX, IncA/C, and IncHI2, play an important part in ESBL gene spread [67]. It is noteworthy that both *E. coli* genomes carried Incl1 plasmid. Our findings show that for Isolate 1143, some AMR genes were harboured in plasmids or other mobile genetic elements, e.g., aminoglycoside resistance genes in the IncR plasmid and the *bla*_TEM-1_ gene (resistance to ampicillins) were found to be carried in the Tn2 transposon (https://transposon.lstmed.ac.uk/tn-registry) (accessed on 1 January 2022). Conversely, for Isolate 1144, the ciprofloxacin resistance gene (*qnrB19*) was carried by Col440I; this is a plasmid recently associated with enterobacteria-carrying ESBL and carbapenemases genes isolated from environment and humans in South Africa [68]. In our analysis, we found that many AMR genes appeared to be inserted in the chromosome—such as *bla*_CTX-M-55_ and *bla*_CTX-M-8_—pointing to the mobility of these genes from mobile genetic elements to the chromosome and vice versa and between different plasmids [69].

Our study has some limitations. The presence of AMR genes harboured in plasmids points to horizontal transfer of AMR determinants. However, conjugation assays should be performed for these isolates to confirm the transfer of genes via plasmids to other bacteria. In addition, isolates were not tested to phenotypically confirm AmpC, pathogenicity, and/or EAST_1_ toxin production. Due to laboratory limitations and restrictions, it was impossible to perform these confirmatory assays. Another limitation is the low number of isolates available for characterisation, which may have limited the scope of detection of important resistance/virulence genes and plasmids.

## 5. Conclusions

Isolates 1143 and 1144 were identified as ESBL-producing *E. coli*, with serotypes -:H46 and O9:H10; phylogroups E and A, and ST/CC 5259/- and 227/10, respectively. They were characterised as potential human pathogens, and they carried multiple virulence genes; Isolate 1143 ST5259 harboured the *astA* gene, encoding the EAST_1_ heat-stable toxin. Both genomes were carriers of a *bla*_EC-15_ gene (AmpC), displaying carbapenem resistance, but not harbouring carbapenemase genes. Other ESBL genes (*bla*_CTX-M-8_ and *bla*_CTX-M-55_) among various AMR genes were detected, mainly located in plasmids or the chromosome. *E. coli* of the ST227 lineage were reported in other countries, while *E. coli* ST5259 reports were rare. Both isolates were found in a remote area in the highlands of Peru, which is of public health concern considering the likely anthropogenic origin derived from incorrect and often unrestricted use of antibiotics in both humans and livestock. Our results can help identify and track *E. coli* strains that pose a risk to human, animal, and environmental health in rural Andean communities. The *E. coli* isolates originated from an animal and DW, highlighting the importance of comprehensive research for preventive action along the One-Health continuum in isolated Andean communities. The sharing of living spaces between humans and animals and the use of contaminated DW could facilitate the transfer of these pathogens to all environs in the community. Thus, new research paths to limit the spread of AMR should focus on the epidemiology of ESBL/AmpC *E. coli*, particularly carbapenem-resistant strains.

## Figures and Tables

**Table 1 antibiotics-11-00692-t001:** Characterisation of the ESBL-producing *E. coli* isolates according to origin and source, sequence type and clonal complex, serotype, virulence genes, resistance phenotype and genotype, and plasmid type.

ID *E. coli* Strains	Source	ST/CC	Serotype	Phylogroup	Virulence Genes	Resistance Phenotype	Resistance Genotype	Plasmid Type
1143	dog faeces	5259/-	-:H46	E	*astA*, *chuA*, *cia*, *gad*, *ompT*, *terC*	SAM, TZP, KZ, CXM, CTX, CAZ, FEP, MEM, SXT	*aac(3)-IId, aph(3*″)*-Ib, aph(6)-Id*, *aadA1*, *aadA2*, *floR*, *bla*_TEM-1_, *bla*_EC-15_, *bla*_CTX-M-8_*, tet*(M), *tet*(B) *dfrA12*	IncI1-IAlpha, IncR, IncFIB(AP001918), IncFIC(FII)
1144	drinking water	227/10	O9:H10	A	*capU*, *cea*, *cia*, *fyuA*, *iutA*, *sitA*, *irp2*, *iroN*, *terC*	SAM, TZP, KZ, CXM, CTX, MEM, CN, CIP	*aph(3*′)*-IIa*, *cmlA1*, *bla*_CTX-M-55_, *bla*_EC-15_, *fosA3*, *qnrB19*, *sul3*, *tet*(B)	IncFIB, Col440I, IncFII, IncI1-IAlpha, IncFII(pHN7A8)

Abbreviations: CC, clonal complex; ST, sequence type. Resistance phenotype: SAM, ampicillin/sulbactam; TZP, piperacillin/tazobactam; KZ, cefazolin; CXM, cefuroxime; CTX, cefotaxime; CAZ, ceftazidime; FEP, cefepime; MEM, meropenem; SXT, trimethoprim/sulfamethoxazole; CN, gentamicin; CIP, ciprofloxacin.

## Data Availability

The data presented in this study are available on request from the corresponding author. The raw reads and assembled data have been submitted to National Center for Biotechnology Information (NCBI) under BioProject Number PRJNA816508.

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
