# Peer review of "Whole-Genome Characterisation of ESBL-Producing E. coli Isolated from Drinking Water and Dog Faeces from Rural Andean Households in Peru"

_antibiotics, 2022, doi:10.3390/antibiotics11050692_

Round 1

Reviewer 1 Report

This study aims to confirm the identity and AMR profile and determine the serotype, phylogroup, sequence type (ST) and clonal complex (CC), pathogenicity, virulence genes, ESBL genes, and type of plasmids for the ESBL-producing E. coli isolates. The paper is interesting, however, the number of the isolates is very low

  1. The number of the collected samples is low.
  2. what is the concentration of the used antibiotics?
  3. please provide more information on the VITEK® 2 Compact automated system used for bacterial identification.
  4. What about statistical analysis?
  5. Line 166: table A1. is this correct?
  6. you need to explain the phylogenetic analysis.
  7. please provide the correlation between phenotypic and genotypic analysis.

Author Response

Reviewer 1

We thank the reviewers for their feedback. We have tried to address the issues raised to the best of our ability.

  1. Comment: The number of the collected samples is low.

Author´s response: thanks for the comment. The reviewer rightly brings up this issue. In our discussion we pointed out that the low number of isolates available for characterization is one of the limitations of our work, because it may have limited the scope of detection of important resistance or virulence genes and plasmids. However, we did not collect the two isolates by direct sampling of the population under study. The two E.coli isolates were obtained as a result of the sampling and analysis carried out by Hartinger et al. [1] in the same study area. They sampled 40 households, with N= 80 for animal faecal samples (companion animals and farm animals) and N= 80 for household´s water samples (drinking water and the household´s primary water source), finding only one ESBL-isolate from a dog and another ESBL-isolate from the drinking water source. Thus, there were no more ESBL-positive isolates available to be characterised.

  1. Comment: what is the concentration of the used antibiotics?

Author´s response: thanks for the comment. The range of concentrations used for the antibiotics sensibility test by the Vitek® 2 system is done in a completely automatic way. Spanu et al. [2] indicate that the VITEK® 2 system is used for routine detection of ESBL production, using a determined ESBL antibiotics´ test panel according to the particular card used. According to the manufacturer [3], “each card contains at least one positive control well with no antibiotic (growth-promoting broth only) and multiple wells with increasing concentrations of various antibiotics in the broth. Growth in the positive control well is monitored until a pre-determined minimum amount of bacterial growth is detected through turbidity measurements. Growth in the control well shows that the test isolate is viable and growing at an appropriate rate to begin analysis of the various drug wells.”

The MICs for each antibiotic in the test panel are determined by comparing the growth of the isolate to the growth of isolates with known MICs, stored in the system´s memory. VITEK® 2 continuously monitors growth over time using multiple parameters, applying a range of antibiotic concentrations (at least three different concentrations per antibiotic).

  1. Comment: please provide more information on the VITEK® 2 Compact automated system used for bacterial identification.

Author´s response: thanks for the comment. The VITEK2 compact system uses colorimetric reagent identification cards that are incubated and interpreted automatically. It uses different reagent cards for different classes of microorganisms. The reagent cards have a number of test substrates that measure a specific metabolic activity. Test data from an unknown organism are compared to the system´s respective database to provide identification [4].

Changes in the manuscript: We are providing more detail about the card used for the identification of the isolates using the VITEK® 2 Compact automated system in lines 116 – 118.

“The microbiological identification and the antimicrobial susceptibility testing were performed using VITEK® 2 Compact automated system (bioMérieux, France). A colorimetric reagent card (GN) to identify the most significant fermenting and non-fermenting Gram-negative bacilli was used according to the manufacturer’s in-structions.”

  1. Comment: What about statistical analysis?

Author´s response: Statistical analysis was not applicable in our case, given the low number (2) of isolates.

  1. Comment: Line 166: table A1. is this correct?

Author´s response: We apologise for the error. This table and other supporting material are now mentioned as Supplementary Materials in the text.

Changes in the manuscript in lines 186-190:

“We compared the VITEK®2 automated readings for antimicrobial susceptibility to the disk diffusion assay results, as shown in Table S1. It is noteworthy that isolate 1144 showed resistance to FEP by the disk diffusion method, contrary to the results obtained using the automated VITEK®2 platform. The VITEK®2 system identified isolate 1143 as an ESBL producer. Isolate 1144, however, was characterised as ESBL negative based on the MIC results, which were interpreted automatically by the system based on the VI-TEK®2 breakpoints (see Table S1)”.

  1. Comment: you need to explain the phylogenetic analysis.

Author´s response: Thanks for bringing up the issue. We did not include the phylogenetic analysis in this version of the manuscript, given it did not provide much relevant information. We could not identify any patterns with the genes analysed because their number was too small.

  1. Comment: please provide the correlation between phenotypic and genotypic analysis.

Author´s response: We thank the reviewer for the comment.

In the Results section, the identity of the isolates and their ESBL-producing capacity was evidenced by the phenotypical analysis, and was confirmed by the genotypic analysis using bioinformatic tools.

Additionally, in Table 1 we show the phenotypic and genotypic resistance profiles for both isolates, and we see correlation between both, except for meropenem. The resistance genes identified as a result of the bioinformatic analysis explain the phenotypical resistance to the antibiotics tested in the phenotypical assay by the VITEK®2 system. For instance, we identified the presence of genes encoding resistance to cephalosporins (blaCTX-M-8, blaCTX-M-55, blaEC-15), and both isolates showed phenotypical resistance to various antimicrobials that belong to this class (cefazolin, cefuroxime, cefotaxime, ceftazidime and cefepime). Further, in one isolate we detected a gene that encodes for broad spectrum β-lactamases (blaTEM-1), and phenotypical ampicillin resistance was also identified (ampicillin/sulbactam). Genes encoding resistance to aminoglycosides [aac(3)-IId, aph(3')-IIa, aph(3'')-Ib, aph(6)-Id, aadA1 and aadA2], quinolones (qnrB19),  trimethoprim (dfrA12) and sulphonamides (sul3) were detected, and phenotypical resistance to gentamycin, ciprofloxacin and to the combined antibiotic trimethoprim/sulfamethoxazole was also found.

Given the information is shown in Table 1, we added a short paragraph to avoid making the manuscript too lengthy.

Changes in the manuscript: (lines 203-205)

“The genotypic resistance of both isolates correlated with their phenotypical resistance, except for ME; the resistance to this carbapenem was only phenotypically observed. “

Reviewer 2 Report

Dear authors,

this paper is a well documented work displaying the increasing risk of antimicrobial resistance even in geographically distant areas, as the Andean mountain range. The manuscript is well written and organized, the methodology is solid and the litterature review is thorough.

Despite the small number of bacterial strains, the study is of great interest as it points out some shortcomings of the VITEK system compared to a thorough molecular analysis (of course the limited numbers do not permit methods comparison) while it gives prominence to the need of implementing Integrated Public Health polices; one of the strains was found in drinking water.

Just few suggestions concerning the paper:

Abstract - Line 23: "and determined serotype, phylogroup..." Determined or determine? please correct it, if this is the case.

Materials and Methods - Line 87

"positive isolates analised..." . May the authors mean "analyzed"? If so, please correct the mistyping error.

Discussion

Lines 241-243: "Notably ... data)" You refer to unpublished data but you could cite another reference presenting similar observations in another rural area. 

Lines 296-298: "In Latin ...regions". Please, cite a reference.

Author Response

Reviewer 2

  1. Comment: Abstract - Line 23: "and determined serotype, phylogroup..." Determined or determine? please correct it, if this is the case.

Author´s response: thanks for the suggestion, we overlooked this typo before submitting the manuscript.

Changes in the manuscript (line 23): it has been changed to “…determine serotype, phylogroup…”

  1. Comment: Materials and Methods - Line 87

"positive isolates analised..." . May the authors mean "analyzed"? If so, please correct the mistyping error.

Author´s response: thanks for the suggestion. We purposedly applied the British English spelling but we overlooked some typos. We checked the whole document for consistency.

Changes in the manuscript (line 89):

” …positive isolates analysed…”

  1. Comment: Discussion

Lines 241-243: "Notably ... data)" You refer to unpublished data but you could cite another reference presenting similar observations in another rural area. 

Author´s response: Thanks for the suggestion. We have included two additional references to back up our group´s unpublished data.

Changes in the manuscript: in lines 267-271 the text was changed to:

“Notably, in our setting, back-yard chicken farming is widespread (83% of households), and farm animals, in general, are administered antibiotics by the owners without pre-scription nor technical supervision (unpublished data from our research group). Other studies carried out in rural locations in Peru show similar findings [46,47].”

  1. Comment: Lines 296-298: "In Latin ...regions". Please, cite a reference.

Author´s response: thanks for the feedback.

Changes in the manuscript: in lines 340-343 the text was changed to:

“According to Guzmán-Blanco et al. [63], in Latin America, the rates of nosocomial infection caused by ESBL-producing enterobacteria, -especially CTX-M enzyme producers-, are higher than in other regions.”

Reviewer 3 Report

Review

Research description:

This study aims to describe the genome sequencing whole-genome characterization of ESBL-producing E. coli isolated from drinking water and dog faeces from rural Andean households in Peru.

The authors should address the following:

Major critics:

The English should be carefully revised by an English speaker, there are paragraphs very difficult to follow.

The study is limited to the rural households of Peru, so it should be generalized to the Andean region, which comprises different countries. The title should be changed removing Andean.

The introduction lacks a storyline. It is very difficult to understand what the main problem authors are aiming to solve.  

Section 2.1 and Section 2.2. The methodology should be carefully revised and focused on this study. The method of sampling is confusing. The authors mentioned that the samples were from domestic dog fasces and water and in this section is mentioned: dogs, cats, pigs, etc; this being the method already described in a previous paper by Hartinger et al., 2021

In section 2.3 the methods used should be briefly described; the only cited paper on previous methods is presented making it difficult to reproduce the experiments. 

Section 2.4. The registered date of the Bio project should be mentioned.

Section 3.1 The number of samples careened for AMR should be mentioned. There is a discrepancy between sections 2.1-2.2 of sampling and results.

Section 3.2. Line 183-184 is confusing

Section: 4.  The results should be discussed in comparison with other similar studies. There is missing literature to be cited.  

Line 265-267. No citation  

The authors claim that “Despite the phenotypic resistance to meropenem of both isolates in our study, we did not find evidence of well-known carbapenemase-encoding genes -blaKPC, blaNDM, blaNDM, 303 blaIMP, blaVIM, and blaOXA-48- or their variants “. In figure 1 several E. coli strains are compared, and the antibiotic resistance genes are shown. The blaOXA181 gene (non- blaOXA48 variant) is shown in the Peru2020, Canada 2016, and US2020. Did the author check the differences between these gene variants?  This should be also discussed as found in other E.coli strains.  

The conclusion section should be rephrased according to the main findings.

Minor critics:

Abstract: define the abbreviation were mentioned first in the document

Keywords: should refer to the main concepts of the research topic; Latin America and rural communities are not keywords, these should be replaced.  

The species, families, order should be written italic

Use synonym for Patio: for example, courtyard

Table A1 should be mentioned as a supplementary Table and cited according to the author's instructions.

Author Response

Reviewer 3

  1. Comment: The English should be carefully revised by an English speaker, there are paragraphs very difficult to follow.

Author´s response: We appreciate the comment. The manuscript has been fully revised again. Paragraphs that were too long and hard to read were divided in two, and punctuation was revised accordingly.

  1. Comment: The study is limited to the rural households of Peru, so it should be generalized to the Andean region, which comprises different countries. The title should be changed removing Andean.

Author´s response: We appreciate the comments. However, we think we cannot generalise the results of a study focused on rural households in the highlands of Peru to the whole Andean region, because we would be assuming that all rural communities in these countries are similar. Peru has three main geographical areas with different characteristics, the Amazon jungle, coastal desserts, and the Andean highlands, and rural populations living in each area show differences in health conditions, environment and resources [5] . Removing the word Andean from the title would imply that all rural households in Peru share the same characteristics and thus, we would be ignoring the marked differences between rural communities from different geographical areas in our country and other countries in the region.

  1. Comment: The introduction lacks a storyline. It is very difficult to understand what the main problem authors are aiming to solve.  

Author´s response: Thanks for the feedback. We aimed at characterising both E. coli isolates at the genomic level, given they represent a serious problem by being ESBL-producers isolated from different sources (animal and drinking water samples from rural households) in a remote setting in a Peru.

In the Introduction, we have tried to emphasise the importance of our study considering the characteristics of most South American countries and the high rates of AMR in the region. This is more relevant due to the scarcity of whole-genome based studies on ESBL-producing enterobacteria in Peru.

Changes in the manuscript:  the text was modified as follows in lines 41-57:

“…Antimicrobial resistance (AMR) -particularly in the Enterobacteriaceae family- has become a problem of great relevance worldwide, due its increasing prevalence and the emergence of multiple drug-resistant strains [1]. AMR can affect everyone, irrespective of age. It is estimated that, in 2019, 4.95 million people died from illnesses associated with bacterial AMR. Of those, 1.27 million deaths, most in low- and middle-income countries (LMICs), were directly attributable to bacterial AMR [1]. Enterobacteria are also the most important etiological agents of serious hospital-acquired and community-onset bacterial infections in humans [2–5]. In South America, resistance to β-lactam antibiotics and fluoroquinolones is a major problem are among the primary obstacles when treating enterobacterial infections, especially in South America [6]. In addition, reports of the emergence of colistin resistance in this region have been recently published [7,8]. South American countries show some of the higher rates of AMR in Enterobacteriaceae worldwide. The huge socioeconomic differences, the broad access to anti-microbials, and the ineffective health systems and sanitation problems favour the emergence and spread of resistant bacterial strains [6]. In this region, antibiotic use is widespread in human medicine and animal production, thus antibiotic residues, and resistant enterobacteria and AMR genes have been detected in soil, water from different environs, agricultural products (produce), and livestock. “

  1. Comment: Section 2.1 and Section 2.2. The methodology should be carefully revised and focused on this study. The method of sampling is confusing. The authors mentioned that the samples were from domestic dog fasces and water and in this section is mentioned: dogs, cats, pigs, etc; this being the method already described in a previous paper by Hartinger et al., 2021

Author´s response: Thanks for the feedback. We did not obtain the two E.coli isolates by direct sampling of the population under study. The two E. coli isolates were obtained as a result of the sampling and analysis carried out by Hartinger et al. in the same study area. They sampled 40 households, with N= 80 for animal faecal samples (companion animals and farm animals: dogs, cats, pigs, fowl, among others) and N= 80 for household´s water samples (drinking water and the household´s primary water source). They found only one ESBL-isolate from a dog and another ESBL-isolate from the drinking water source, and they were used in our study for characterization.

Changes in the manuscript:  the text was modified as follows:

2.1. Study setting and design.

 Hartinger et al. studied the dissemination pathways of AMR in humans, animals, and the environment in the Cajamarca Region in the northern highlands of Peru. The cross-sectional study design and sampling scheme are described elsewhere [23]. In brief, they sampled 40 households. Their study collected two rectal or cloacal swabs of fresh stool samples from a domestic animal (dog, cat) and/or a farm animal (cow, pig, fowl) from each household (N=80). Additionally, two water samples per household were collected, one from the household’s drinking water (DW) source and/or one from the household’s primary water source (N=80). The two ESBL-positive isolates analysed in the present study originated from two out of 40 households sampled. They were isolated from one water sample and one dog faeces sample, as shown in Figure S1. 

2.2. Households’ characteristics

Hartinger et al. found only one ESBL-isolate from a dog and an ESBL-isolate from the DW source from two homes [23]. In these households, animals were allowed to roam freely in the kitchen and courtyard, and their faeces were found in both areas. Both homes used piped water from the public water network system but had no sewage system, as they had installed latrines instead.

  1. Comment: In section 2.3 the methods used should be briefly described; the only cited paper on previous methods is presented making it difficult to reproduce the experiments

Author´s response: We thank the reviewer for his comment. We described very briefly the methods used by Hartinger et al. and in addition, we made a figure (Figure S1) for better understanding.

Changes in the manuscript:  the text was modified as follows:

“2.43. Phenotypical and molecular identification of ESBL-producing ability

The phenotypical ESBL-producing ability of the isolates was determined using the Jarlier method [28] for the following antibiotics: aztreonam (5-µg disk), ceftazidime (30-µg disk), cefotaxime (30-µg disk), ceftriaxone (30-µg disk), amoxicillin-clavulanic acid (30-µg disk) and cefepime (30-µg disk), and confirmed by the combined disc method [27].”

.”

  1. Comment: Section 2.4. The registered date of the Bio project should be mentioned.

 Author´s response: We thank the reviewer for his comment. However, it is not a standard practice to include the registry date, given it is visible when the Bioproject is accessed using the number provided in the manuscript.

  1. Comment: Section 3.1 The number of samples screened for AMR should be mentioned. There is a discrepancy between sections 2.1-2.2 of sampling and results.

Author´s response: Thanks for the suggestion. Probably the way it was written was not clear for the reviewers. We have tried to clarify this issue with the help of Figure S1 and changing the text for those sections. Please see our response to Reviewer 3 comment 4.

  1. Comment: Section 3.2. Line 183-184 is confusing

Author´s response: We thank the reviewer for his feedback. We assume the reviewer referred to section 3.1, lines 183-184, which lacked clarity.

Changes in the manuscript:  the text was modified as follows in lines 209-214:

“Genomic analysis using MLST 2.0 indicated that the E. coli isolates 1143 and 1144 belonged to the ST5259 and ST227, respectively, and 1144 belonged to the CC 10. Ac-cording to the genomes´ analysis using the Clermont.typing tool, the 1143 and 1144 isolates corresponded to the phylogenetic groups E and A, respectively. Using SerotypeFinder v.2.0, isolate 1143 was classified as serotype -:H46, whereas isolate 1144 be-longed to serotype O9:H10, as presented in Table 1. “

  1. Section: 4.  The results should be discussed in comparison with other similar studies. There is missing literature to be cited.  

Author´s response: We thank the reviewer for the comment. We have used various additional references in the discussion.

  1. Comment: Line 265-267. No citation  

Author´s response: We thank the reviewer for the feedback.

The paragraph in which the reviewer finds no citation was not written based on other publications. It is a result of direct observation and interpretation of the output of the application of the Virulencefinder tool. The results of the bioinformatic analysis are reported as supplementary materials, Table S3.

Changes in the manuscript: in lines 304-307 the text was modified as follows:

“Many of the identified virulence genes were associated with iron acquisition systems, and a small part was associated with toxin production, metal resistance (tellurite), and other virulence determinants, as shown in Table S3.”

  1. Comment: The authors claim that “Despite the phenotypic resistance to meropenem of both isolates in our study, we did not find evidence of well-known carbapenemase-encoding genes -blaKPC, blaNDM, blaNDM, 303 blaIMP, blaVIM, and blaOXA-48- or their variants “. In figure 1 several E. coli strains are compared, and the antibiotic resistance genes are shown. The blaOXA181 gene (non- blaOXA48 variant) is shown in the Peru2020, Canada 2016, and US2020. Did the author check the differences between these gene variants?  This should be also discussed as found in other colistrains.

Author´s response: We thank the reviewer for the feedback. We did not include the phylogenetic analysis in this version of the manuscript. Please see the reasons for our decision in the response to Reviewer 1, Comment 6. We have discussed other E. coli strains that showed carbapenemase-encoding genes reported in other parts of the world.

Changes in the manuscript:  in lines 286-289 the text was modified as follows:

“…In Tunisia, an ST227 carbapenem-resistant E. coli isolate from human origin was reported in 2016, carrying the blaOXA-48, blaCTX-M-15, blaTEM, and blaOXA-1 genes [49]. A carbapenem- resistant E. coli isolate of the ST227 lineage was detected in Lebanon, carrying the blaOXA-48 and the blaTEM-1 genes.”

  1. Comment: The conclusion section should be rephrased according to the main findings.

Author´s response: We thank the reviewer for the feedback. We have rephrased the conclusions accordingly.

Changes in the manuscript:  the conclusion was modified as follows:

 “The isolates 1143 and 1144 were identified as ESBL-producing E. coli, with serotypes -:H46 and O9:H10; phylogroups E and A, and ST/CC 5259/- and 227/10, respectively. They were characterised as potential human pathogens, and they carried multiple virulence genes; isolate 1143 ST5259 harboured the astA gene, encoding the EAST1 heat-stable toxin. Both genomes were carriers of a blaEC-15 gene (AmpC), displaying carbapenem-resistance but not harbouring carbapenemases genes. Other ESBL genes (blaCTX-M-8 and blaCTX -M-55) among various AMR genes were detected, mainly located in plasmids or the chromosome. E. coli of the ST227 lineage were reported in other countries, while E. coli ST5259 reports were rare. Both isolates were found in a remote area in the highlands of Peru, which is of public health concern considering the likely anthropogenic origin derived from incorrect and often unrestricted use of antibiotics in both humans and livestock. Our results can help identify and track E. coli strains that pose a risk to human, animal, and environmental health in rural Andean communities. The E. coli isolates originated from an animal and DW, highlighting the importance of comprehensive research for preventive action along the One-Health continuum in isolated Andean communities. The sharing of living spaces between humans and animals and the use of contaminated DW could facilitate the transfer of these pathogens to all environs in the community. Thus, new research paths to limit the spread of AMR should focus on the epidemiology of ESBL/AmpC E. coli, particularly carbapenem-resistant strains. ”

  1. Abstract: define the abbreviation were mentioned first in the document.

Author´s response: Thanks for the suggestion. We revised and made sure that all the abbreviations used in the abstract and in the text have been defined when used for the first time.

Changes in the manuscript:  the text was modified as follows:

“Abstract: E. coli that produce extended-spectrum β-lactamases (ESBL) are major multidrug-resistant bacteria.”

  1. Comment: Keywords: should refer to the main concepts of the research topic; Latin America and rural communities are not keywords; these should be replaced.  

Author´s response: We thank the reviewer for the feedback. We made changes to the keywords, according to the journal´s instructions.

Changes in the manuscript:  the text was modified as follows:

Keywords: phylogenomic analysis; One Health; ESBL-producing E. coli; carbapenem resistance; whole-genome-sequencing

  1. Comment: The species, families, order should be written italic

Author´s response: Thanks for the feedback, we apologise for overlooking this issue. The manuscript has been checked thoroughly and the corresponding changes have been implemented.

Changes in the manuscript:

Lines:41-43

“…Antimicrobial resistance (AMR) -particularly in the Enterobacteriaceae family- has become a problem of great relevance worldwide, due its increasing prevalence and the emergence of multiple drug-resistant strains.”

Lines:51-52

“…South American countries show some of the higher rates of AMR in Enterobacteriaceae worldwide.”

Lines: 319-321

“…The blaCTX-M-55 is one of the most abundant ESBL genes in the Enterobacteriaceae family, with a rising prevalence in E. coli from livestock and humans, especially in China [54,55].”

  1. Comment: Use synonym for Patio

Author´s response: thanks, we have changed patio by courtyard.

Changes in the manuscript: in lines 109-110 the text was modified as follows:

“In these households, animals were allowed to roam freely in the kitchen and courtyard, and their faeces were found in both areas. Both homes used piped water from the public water network system but had no sewage system, as they had installed latrines instead. “

  1. Comment: Table A1 should be mentioned as a supplementary Table and cited according to the author's instructions.

Author´s response: Thank you for the feedback. This reporting mistake was corrected. Please see our response to reviewer 1, comment 5.

Reviewer 4 Report

Maria L. Medina-Pizzal and colleagues (antibiotics-1687560) use whole genomic sequencing approach and some well-established online toolkits for studying Two E. coli recovering from Peru, where this type of study may be very limited. In general, this study is very simple, with most of work are in silico analysis, and one type of AMR experiment. Additionally, Lots of vague points are there, and this study can be improved in certain aspects.

  1. “The isolates were pathogenic,” how did you known that they are pathogenic? Did you conduct some animal infection studies? In silico Virulence Factor analysis is not enough?
  2. “not showing 33 a close genetic relationship to them” what is your definition for "close" relationship?
  3. “However, few reports characterized ESBL-producing 67 enterobacteria molecularly 

”, are these English, I assume you mean in molecular level. Please check it throughout the manuscript.

  1. Method 2.1 and 2.2 is the repeat for previous study, can you make some informative figures or tables, as the supplemental stuffs.
  2. Method 2.5. detailed excel information for 84 ref. genomes are needed, we want all the detail for reproducible work… Please make all data needed.
  3. Have you checked the chromosome based mutations for AMR?
  4. You show three different in silico method for AMR detection, and you get the difference, so what? Any discussion about these differences?
  5. All the detailed results for in silico analysis, including VFs, should be showed as the table or supplemental table
  6. You did some phylogeny analysis with ST227, but not for ST5259, you might explain do the similar analysis?
  7. The relevant literature is limited, so please include the following most recent ones.

doi: 10.1089/mdr.2016.0090.

https://pubmed.ncbi.nlm.nih.gov/?term=Escherichia+coli+ST5259+genome%5BTitle%2FAbstract%5D&sort=date&size=200

doi:10.3390/microorganisms7100461.

doi: 10.3390/antibiotics9020080

Author Response

Reviewer 4

  1. Comment: “The isolates were pathogenic,” how did you known that they are pathogenic? Did you conduct some animal infection studies? In silico Virulence Factor analysis is not enough?

Author´s response: Thanks for the feedback. The reviewer rightly brings up this issue, which we overlooked. In silico analysis is used a predictive model; however, results from this kind of analysis must be confirmed by in vitro and in vivo assays.

Changes in the manuscript:  the text was modified as follows:

Abstract:

“… The isolates were ESBL-producing, and carbapenem-resistant, not harbouring carbapenemase-encoding genes. Isolate 1143 ST5259, harboured the astA gene, encoding the EAST1 heat-stable toxin. Both genomes carried ESBL genes (blaEC-15, blaCTX-M-8, and blaCTX -M-55). Nine plasmids were detected, namely IncR, IncFIC(FII), IncI, IncFIB(AP001918), Col(pHAD28), IncFII, IncFII(pHN7A8), IncI1, and IncFIB(AP001918). Finding these potentially pathogenic bacteria is worrisome given their sources, and highlights the importance of One-Health research efforts in remote Andean communities. “

In section 3.2:

Lines: 217-218

“Using Pathogenfinder 1.1 and VirulenceFinder 2.0, both isolates were identified as likely human pathogens, but Shiga-toxin genes were not detected in their genomes.”

In Discussion:

Lines: 263-265

“… Thus, finding an ESBL-producing, potentially pathogenic, phylogroup-E isolate in dog faeces points to a probable transmission route from humans.”

Lines: 380-381

“… In addition, isolates were not tested to phenotypically confirm AmpC, pathogenicity, and/or EAST1 toxin production. “

  1. Comment: “not showing a close genetic relationship to them” what is your definition for "close" relationship?

Author´s response: Thanks for bringing up the observation. This observation was removed from the manuscript as it referred to the phylogenetic analysis which was also removed. The reasons for not including this analysis are explained in the response to Reviewer 1 Comment 6.

  1. Comment: “However, few reports characterized ESBL-producing enterobacteria molecularly”, are these English, I assume you mean in molecular level. Please check it throughout the manuscript.

Author´s response: Thanks for the suggestion. We have checked the English usage throughout the manuscript.

Changes in the manuscript in lines 78-82:

“In Peru, previous studies investigated ESBL in enterobacteria isolated from clinical, community, and peri-urban settings. However, few reports characterised ESBL-producing enterobacteria at the molecular level, and if so, mostly in clinical settings [18–22]. Using molecular methods, we previously characterised ESBL-producing enterobacteria from rural Andean communities.”

  1. Comment: Method 2.1 and 2.2 is the repeat for previous study, can you make some informative figures or tables, as the supplemental stuffs.

Author´s response: Thanks for the suggestion. We have made a figure (Figure S1) that illustrates the sampling done by Hartinger et al. and how this leads to our study.

  1. Comment: Methods 2.5. detailed excel information for 84 ref. genomes are needed, we want all the detail for reproducible work… Please make all data needed.

Author´s response: Thanks for the comment. Given that we did not include the comparative phylogenomic analysis in this version, a table containing the IDs and other relevant information for the 84 genomes is not needed either.

  1. Comment: Have you checked the chromosome-based mutations for AMR?

Author´s response: Thanks for the comment. Given our specific aim, we did not look for this additional information.  We understand that point mutations in specific chromosomal genes can cause resistance. However, we were interested in determining whole ESBL genes (among other AMR genes) and we accomplished this by setting identity at ≥ 90% and coverage at ≥ 90% during the application of the bioinformatic tools. Most genes found showed a perfect identification match and many of them were found in plasmids.

  1. Comment: You show three different in silico method for AMR detection, and you get the difference, so what? Any discussion about these differences?

Author´s response: We thank the author for bringing up this issue. We used ResFinder and NCBI databases for AMR genes detection and the information was essentially the same. However, these tools did not allow us to identify if the genes were located in a plasmid (or another mobile element) or in the chromosome. Therefore, we used the MobileElementFinder v.1.0 tool to distinguish their location.

  1. Comment: All the detailed results for in silico analysis, including VFs, should be showed as the table or supplemental table.

Author´s response: Thanks for the comment.  The outputs of the bioinformatic tools used have been included under Supplementary Materials.

“Supplementary Materials: The following supporting information can be downloaded at: www.mdpi.com/xxx/s1, Figure S1: Obtention of the E. coli isolates 1143 and 1144; Table S1: Antibiotic susceptibilitya profile for E. coli isolates 1143 and 1144 determined by the VITEK®2 system and compared to the disk diffusion assay resultsb;Table S2: Genomes used for comparison in phylogenomic analysis; Table S3: Outputs of bioinformatic tools.”

.”

  1. Comment: You did some phylogeny analysis with ST227, but not for ST5259, you might explain do the similar analysis?

Author´s response: Thanks for the comment. Please see our response to Comment 5.

  1. Comment: The relevant literature is limited, so please include the following most recent ones.

doi: 10.1089/mdr.2016.0090. (KPC-2-Producing Escherichia coli, mentions ST227)

https://pubmed.ncbi.nlm.nih.gov/?term=Escherichia+coli+ST5259+genome%5BTitle%2FAbstract%5D&sort=date&size=200 (search in Pubmed zero results)

doi:10.3390/microorganisms7100461 (colistin resistance, systematic review and meta-anlaysis)

doi: 10.3390/antibiotics9020080 (about colistin resistance, mcr-1 gene)

Author´s response: Thanks for the feedback and the suggested literature. We added some publications in relation to the emergence of colistin resistance in the Latin America region under Introduction and Discusion. We also discussed other publications on carbapenem resistance and specific bla genes and their carriage by plasmids.

Changes in the text:

                             Lines 48-51

“…In South America, resistance to β-lactam antibiotics and fluoroquinolones is a major problem when treating enterobacterial infections [6]. In addition, reports of the emergence of colistin resistance in this region have been recently published [7,8].”

Lines 348-350

“…Fortunately, more than 90% of ESBL-producing enterobacteria are still susceptible to carbapenems [63], although reports of carbapenem-resistant enterobacteria alarmingly increase in Latin American countries [65].”

Lines 286-289

“In Tunisia, an ST227 carbapenem-resistant E. coli isolate from human origin was reported in 2016, carrying the blaOXA-48, blaCTX-M-15, blaTEM, and blaOXA-1 genes [49]. A carbapenem- resistant E. coli isolate of the ST227 lineage was isolated from clinical samples in Lebanon, carrying the blaOXA-48 and the blaTEM-1 genes [50]. ”

Lines 319-323

“The blaCTX-M-55 is one of the most abundant ESBL genes in the Enterobacteriaceae family, with a rising prevalence in E. coli from livestock and humans, especially in China [55,56]. It is usually carried by plasmids, but it has also been found chromosomally [56]. In Canada, enterobacteria isolated from turkeys carried the blaCTX-M-55 gene mediated by IncF plasmids [57].”

Round 2

Reviewer 1 Report

I still have concerns about the number of the samples and the statistical analysis

Author Response

Rebuttal Letter

We have tried to address the comments from the reviewers in the best way possible.

Reviewer1

  1. Comment: I still have concerns about the number of the samples and the statistical analysis

Author´s response:

Thanks for the comment. We agree that having more samples would have been ideal; however, we had only two ESBL isolates for characterization in this study. As mentioned in our previous response, Hartinger et al.[1] only collected samples from 40 households. They collected 80 samples from animal stools (companion animals and farm animals) and 80 samples from each household´s water (drinking water and its primary water source). (Additionally, they also collected samples from children’s stools (N=40), courtyard soil (N=40), and the community´s reservoirs water (N=26), giving a total of 266 samples). This information is shown in Supplementary Materials Figure S1.

For all samples, the prevalence of MDR profiles was nearly 9 %, and only two enterobacteria isolates were ESBL-positive, one belonging to a dog stool and another to a drinking water source [1].

We decided to use only the two ESBL-positive isolates available to be characterized for this study. Statistical analysis was not applicable in this case, given the limited number of isolates.

In addition, we have recognized and discussed the limited scope of detection of significant resistance or virulence genes and plasmids due to the limited number of available ESBL isolates. However, we believe that the finding of potentially pathogenic ESBL-producing and carbapenem-resistant E. coli in this particular location is what stands out the most in this study. Further, the isolates were obtained from drinking water and a companion animal from households in a rural Andean community in Peru, where poverty is prevalent and health services are not easily accessible [2].

Reviewer 3 Report

The authors improved the manuscript but there are still some concerns about the sampling. The number of samples is too low in comparison with previous studies. 

At section 3.2 it was mentioned that both isolates are “likely human pathogens” based on the VirulenceFinder and PathogenFinder without presenting any values that are given by each tool. Some data are presented in TableS2, and the authors do not make any reference to these results. Same for the other tools. 

Complementary experimental analysis should be considered to confirm the in-silico results. 

Author Response

Reviewer 3

We have tried to address the comments from the reviewers in the best way possible.

Comments and Suggestions for Authors

  1. Comment: The authors improved the manuscript but there are still some concerns about the sampling. The number of samples is too low in comparison with previous studies.

Author´s response:  Thanks for the comment. Please read our response to the Reviewer 1.

  1. Comment: At section 3.2 it was mentioned that both isolates are “likely human pathogens” based on the VirulenceFinder and PathogenFinder without presenting any values that are given by each tool. Some data are presented in TableS2, and the authors do not make any reference to these results. Same for the other tools. 

Author´s response: We apologise for the oversight, we have made the corresponding references to Supplementary Materials Table S2 in the text. All the outputs for the bioinformatic tools are available in Table S2. Further, all the relevant virulence genes and other data obtained from these outputs are listed in Table 1. For the sake of clarity, we reorganized some of these outputs and provided a short summary inside Table S2, for Pathogenfinder and Virulencefinder as shown below:

For Pathogenfinder

Isolate 1143 was reported as a human pathogen, with a probability of 0.917.

Isolate 1144 was reported as a human pathogen, with a probability of 0.927.

For Virulencefinder

Summary isolate 1143

Virulence factor

Protein function

astA                   

EAST-1 heat-stable toxin

chuA                    

Outer membrane hemin receptor

cia                   

Colicin ia

gad                     

 Glutamate decarboxylase

ompT                   

Outer membrane protease (protein protease 7) 

terC                   

Tellurium ion resistance protein

traT                  

Outer membrane protein complement resistance

Shiga-toxin genes

No hit found

Summary isolate 1144

Virulence factor

Protein function

capU

Hexosyltransferase homolog

cea

 Colicin E1  

cia

Colicin ia     

cvaC 

Microcin C 

etsC

Putative type I secretion outer membrane protein

fyuA

Siderophore receptor 

hlyF

 Hemolysin F

iroN   

Enterobactin siderophore receptor protein

irp2

High molecular weight protein 2 non-ribosomal

iss 

 Increased serum survival  

iucC

Aerobactin synthetase

iutA  

 Ferric aerobactin receptor 

mchF  

ABC transporter protein MchF    

ompT  

 Outer membrane protease (protein protease 7)

sitA 

Iron transport protein

terC 

 Tellurium ion resistance protein

traT 

Outer membrane protein complement resistance

Changes in the manuscript: lines 159-160

“The complete outputs for all the results of the bioinformatic tools are shown in Supplementary Materials Table S2.”

  1. Comment: Complementary experimental analysis should be considered to confirm the in-silico results. 

Author´s response: We thank the reviewer for the comment. We agree that ideally, all in silico results should be confirmed by experimental analysis, given the predictive nature of all in silico results. Sadly, we faced laboratory limitations, so it was impossible to address this issue, as the reviewer recommends. Further, some experimental assays, such as the pathogenicity assays, require animal models and/or cellular cultures, which we cannot perform due to the lack of installed capacity in our laboratory in Peru. In lines 379-284, we have added a sentence that refers to the laboratory restrictions/limitations and the impossibility for us to do the complementary experimental assays.

Changes in the manuscript:

“Our study has some limitations. The presence of AMR genes harboured in plasmids points to horizontal transfer of AMR determinants. However, conjugation assays should be performed for these isolates to confirm the transfer of genes via plasmids to other bacteria. In addition, isolates were not tested to phenotypically confirm AmpC, pathogenicity, and/or EAST1 toxin production. Due to laboratory limitations and restrictions, it was impossible to perform these confirmatory assays.”

Reviewer 4 Report

none

Author Response

Thank you for your comment regarding our manuscript.